# Ovicidal Toxicity and Morphological Changes in Housefly Eggs Induced by the Essential Oils of Star Anise and Lemongrass and Their Main Constituents

**DOI:** 10.3390/insects15070481

**Published:** 2024-06-27

**Authors:** Hataichanok Passara, Sirawut Sittichok, Jirisuda Sinthusiri, Tanapoom Moungthipmalai, Cheepchanok Puwanard, Kouhei Murata, Mayura Soonwera

**Affiliations:** 1Office of Administrative Interdisciplinary Program on Agricultural Technology, School of Agricultural Technology, King Mongkut’s Institute of Technology Ladkrabang, Ladkrabang, Bangkok 10520, Thailand; hataichanok.pa@kmitl.ac.th (H.P.); best_pest22@hotmail.com (S.S.); 2Department of Plant Production Technology, School of Agricultural Technology, King Mongkut’s Institute of Technology Ladkrabang, Ladkrabang, Bangkok 10520, Thailand; 64604012@kmitl.ac.th (T.M.); 63604010@kmitl.ac.th (C.P.); 3Community Public Health Program, Faculty of Public and Environmental Health, Huachiew Chalermprakiet University, Samut Prakan 10540, Thailand; jiri_ja@yahoo.com; 4School of Agriculture, Tokai University, Kumamoto 862-8652, Japan; kmurata@agri.u-tokai.ac.jp

**Keywords:** housefly, ovicidal toxicity, ultrastructural changes, lemongrass essential oil, star anise essential oil, geranial, *trans*-anethole, non-target aquatic predator

## Abstract

**Simple Summary:**

Houseflies (*Musca domestica*: Diptera) are serious medical and veterinary vectors of several human and animal pathogens. Ovicides reduce breeding housefly populations and the risk of housefly vector diseases. The natural ovicides of plant essential oils (EOs) and their main ingredients were suitable and sustainable options at this time against houseflies. This study evaluated the housefly ovicidal activities of four single-component formulations and two combination formulations of essential oils and their main constituents. The efficacy of each treatment was compared to that of α-cypermethrin (a synthetic insecticide). Two combinations: lemongrass (*Cymbopogon citratus*) EO + *trans*-anethole and star anise (*Illicium verum*) EO + geranial, were at least 1.1 times as effective in ovicidal activity as all single-component treatments and α-cypermethrin. They induced embryonic damage and mortality inside the activities of the eggs, disrupting the eggshell, hatching line, aeropyles, plastron, and micropyle. More significantly, the two combinations and all single-component formulations were safe for two non-target aquatic species: guppy (*Poecilia reticulate*: Poeciliidae) and molly (*Poecilia latipinna*: Poeciliidae), while α-cypermethrin was not safe for them. The two combinations could be developed into an effective and safe natural ovicide for reducing housefly populations and managing housefly vector diseases.

**Abstract:**

This study attempted to evaluate the ovicidal activity of single-component formulations and combination formulations of lemongrass and star anise essential oils (EOs) and their main constituents against housefly eggs. The efficacies of the combinations were compared with those of single-component formulations and α-cypermethrin. Safety bioassays of all treatments and α-cypermethrin on non-target predators—guppy and molly—were conducted. Two combinations: 1% lemongrass EO + 1% *trans*-anethole and 1% star anise EO + 1% geranial, exhibited a strong ovicidal activity with an inhibition rate of 94.4 to 96.2%. They were 1.1 times as effective as α-cypermethrin. The two combinations also showed high synergistic activity compared to single-component formulations, with a high synergistic index and a high increased inhibition value of 37.4 to 57.7%. All EO treatments were benign for all non-target aquatic species with a high 50% lethal time (LT_50_) and safety index. In contrast, α-cypermethrin was highly toxic to them with a low LT_50_. The morphological abnormalities observed in housefly eggs at death were those such as the shrivelling of the eggs, aberrations and damage to the eggshells, hatching lines, aeropyles, plastron, and micropyle. The potential of these two combinations as a cypermethrin replacement is compelling.

## 1. Introduction

Climate change and global warming are good for the development and prevalence of insect pests, including the housefly *Musca domestica* L. [1,2]. Houseflies are an economic insect pest that exerts a negative impact on the welfare of humans and animals. Typically, flies feed on animal manure, faeces, carrion, and other decaying organic substances. They reproduce by laying eggs on the surface of organic substances [3,4]. The housefly is a mechanical vector of more than 200 human and animal pathogens [4]. Houseflies can spread diseases from animals to animals and from animals to humans [5]. For example, it is a serious vector of poultry avian influenza, and it can transmit the disease from poultry to poultry and from poultry to humans [6]. The annual economic loss from housefly control in agricultural farms and urban areas in the United States was estimated at more than USD 1.87 billion [7,8,9]. Synthetic insecticides have been recommended to reduce and manage housefly populations in outbreak areas [8,10]. Unfortunately, the long-term application of synthetic insecticides has led to the development of houseflies resistant to them [10,11]. The resistance of houseflies to various groups of synthetic insecticides has been reported in several countries: China [10], the United States [11], Pakistan [12], Russia [13], Saudi Arabia [14], and Thailand [15]. Moreover, most synthetic insecticides exert highly negative impacts on human and non-target species, disrupting natural enemies of the insects and pollinators [16,17,18]. 

These global problems can be rectified with green plant insecticides [18]. Several plants have developed phytochemicals for self-defence against pathogens and their insect pests. Among phytochemicals, plant essential oils (EO) are the best substance as a green alternative to synthetic insecticides for the management of pest insects [18]. They are highly effective, exert multiple modes of action, and are benign to humans, mammals, and non-target aquatic predators [19,20]. They are also biodegradable and highly species-specific, making them the best substance for housefly management [20,21].

Presently, a successful strategy for housefly management is to disrupt breeding sites with oviposition deterrent agents and ovicidal agents, reducing their population density [21,22]. Several reports have shown that EOs from many plants provided oviposition deterrent and ovicidal activities against houseflies: eucalyptus (*Eucalyptus globulus*, *E*. *camaldulensis*), lemongrass (*Cymbopogon citratus*), lavender (*Lavandula angustifolia*), neem (*Azadirachta indica*), peppermint (*Mentha piperita*), phlai (*Zingiber cussumunar*), star anise (*Illicium verum*), sweet orange (*Citrus sinensis*), yellow oleander (*Thevetia peruviana*) [21,22], clove (*Syzygium aromaticum*), citronella grass (*C*. *nardus*), and ylang-ylang (*Cananga odorata*) [23]. Five EOs from Umbelliferae plants: ajwain (*Trachyspermum ammi*), sweet fennel (*Foeniculum vulgare*), anise (*Pimpinella anisum*), dill (*Anethum graveolens*), lark daisy (*Centraterum anthelminticum*), and sweet fennel (*F*. *vulgare*) exhibited a highly effective oviposition deterrent against housefly females [24].

However, reports of EO combinations are limited on their ovicidal and oviposition deterrent activities against the housefly. Some researchers have reported the activities of EO constituents against the eggs of human head louse (*Pediculus humanus capitis*: Pediculidae) such as nerolidol, thymol, geranial, carveol, α-terpineol, and methanol [25]. Combinations of EOs: Javanese turmeric (*Curcuma xanthorrhiza*) + eucalyptus (*Eucalyptus globbulus*), zedory (*C*. *zedoaria*) + eucalyptus (*E*. *globulus)*, and wild ginger (*Z*. *zerumbet*) + eucalyptus (*E*. *globulus)*—showed a high ovicidal activity against the eggs of human head lice [26]. A combination of star anise EO + cinnamon (*Cinamomum verum*) EO exhibited highly repellent and ovicidal activities against the American cockroach, *Periplaneta americana* [27]. D-limonene, geranial, *trans*-cinnamaldehyde, and combinations of cinnamon EO (*C*. *verum*) + geranial, lemongrass EO + D-limonene, sour orange (*C*. *aurantium*) + geranial, D-limonene + geranial, geranial + *trans*-cinnamaldehyde, D-limonene + *trans*-cinnamaldehyde all showed ovicidal activities against the eggs of two mosquito vectors (*Aedes aegypti* and *Aedes albopictus*) [20]. They were also benign to two non-target aquatic predators (molly, *Poecilia latipinna,* and guppy, *Poecilia reticulate*) [20].

EOs and their constituents exert their mode of action at different sites in the insect eggs. An extract of *Cipadessa baccifera* (Meliaceae) destroyed the egg membrane of the *Culex quinquefasciatus* mosquito [28]. Curcumin disrupted the egg chitinisation process and embryo development of the *Ae*. *aegypti* mosquito [29]. Lectin from moringa (*Moringa oleifera*: Moringaceae) and *Myracrodruon urundeva* (Anacardiaceae) caused abnormalities in the exocorionic cells of the eggs of *Ae*. *aegypti* [30]. Phenylpropanoids and ketones from clove (*Synzygium aromaticum*) disrupted the embryonic respiratory system of the eggs of dark-winged fungus gnat (*Bradysia procera*; Diptera: Sciaridae) [31]. Additional studies analysed the mechanism of action of EOs and their main constituents against housefly larvae and mosquito eggs from images taken by scanning electron microscopy (SEM) [20,32]. The insecticidal activities of lemongrass and star anise EOs against houseflies have been verified [21,23]. However, for them to be appropriate for controlling large housefly populations, their efficacy against the eggs needs to be verified.

To bridge this gap, this study investigated the ovicidal activities against housefly eggs of several single-component and combination formulations of lemongrass and star anise EOs and their main active constituents. The two EOs and their main constituents have long been used as native medicines and food ingredients in many Asian countries, including Thailand, since ancient times [33,34]. Evaluations were performed on the synergistic ovicidal effects and biosafety of the combinations against non-target aquatic species. The two aquatic non-target species were molly (*Poecilia latipinna* Lesueur: Poeciliidae) and guppy (*Poecilia reticulata* Peters: Poeciliidae). These two non-target species were common aquatic predators in Asia, including Thailand [20]. Ultrastructural changes, underlying the mode of action, were observed under SEM in housefly eggs induced by the single-component and combination formulations. The investigation into the ovicidal activity of these compounds and their safety against non-target organisms, in addition to their reported insecticidal activity in the literature, should demonstrate that they are a sustainable and safe option, worthy of further field testing, for reducing breeding housefly populations and general housefly management, especially in heavily infested areas.

Our contribution to the field of pest insect management is a demonstration of the great potential against housefly eggs of two EO combinations that are safer and more effective than a popular synthetic insecticide, α-cypermethrin.

## 2. Materials and Methods

### 2.1. Essential Oils and Other Chemicals

The lemongrass and star anise EOs were purchased from Chemipan Corporation Company Limited, Kanna Yao, Bangkok, Thailand. The technical grade 96% geranial (CAS 5392-40-5, the main compound of lemongrass EO) and 98.5% *trans*-anethole (CAS 4180-23-8, the main compound of star anise EO) were purchased from Sigma-Aldrich Company Limited, Saint Louis, MO, USA. The two compounds were prepared with 70% *v/v* stock solutions in ethyl alcohol (purchased from Siribuncha Company Limited, Phra Khanong, Bangkok, Thailand). The 10% *w/v* α-cypermethrin (Dethroid 10^®^) used as a positive control was purchased from Penta Chemi Company Limited, Khlong San, Bangkok, Thailand. The drinking water (Kaesad drinking water^®^) used as a negative control was purchased from the School of Food Industry, King Mongkut’s Institute of Technology Ladkrabang (KMITL), Bangkok, Thailand.

### 2.2. The Treatments

Based on our previous studies [35,36,37], the concentrations of four single-component treatments of lemongrass EO at 1.0%, star anise EO at 1.0%, geranial at 1.0%, and *trans*-anethole at 1.0%, and two combination treatments of 1.0% lemongrass EO + 1.0% *trans*-anethole and 1.0% star anise EO + 1.0% geranial were appropriate for evaluating the ovicidal activity against housefly eggs. At these assigned concentrations, these EO treatments have already been shown to be repellent and insecticidal against housefly adults [35,36,37] and ovicidal against mosquito eggs [20]. The solvent for diluting all treatments was 70% *v/v* ethyl alcohol.

### 2.3. Housefly Rearing

The eggs of the housefly *M*. *domestica* were obtained from a housefly colony raised in the entomological laboratory of the School of Agricultural Technology, KMITL, under conditions of 25.8 ± 1.3 °C, 71.5 ± 2.5% RH and 12.5 h light and 11.5 h dark periods. They were reared with steamed mackerel mixed with milk powder in a ratio of 1:0.25 [35,36]. After 1–2 days, the eggs developed into larvae, pupae, and then adults. The adults were fed a 10% honey solution + milk powder + mineral water in a ratio of 5:5:90. The 2–3-day female adults laid their eggs on steamed mackerel. New eggs were collected at 2–3 h to serve as test subjects for the ovicidal bioassay [21] (see the egg life cycle in Figure 1).

### 2.4. Ovicidal Bioassay

A topical application method was used to evaluate the ovicidal efficacy of seven treatments against housefly eggs [21]. Fifteen grams of steamed mackerel were placed on a 5 cm Petri dish, and ten fertile eggs were placed on the steamed mackerel. A volume of 100 µL of each treatment was dropped onto the eggs. The treated eggs were incubated under normal maintenance conditions for housefly eggs at 25.8 ± 1.5 °C, 70.5 ± 1.5% RH, and 12.5 h light and 11.5 h dark periods. Two control treatments with 1% *w/v* α-cypermethrin (positive control) and drinking water (negative control) were arranged concurrently with the treatments of every single-component and combination formulation. Each treatment was replicated five times. The eggs in each treatment were exposed to the treatment for 24 h, checked under a stereomicroscope, then exposed to the treatment further for 48 h and checked whether they hatched or did not hatch. Egg mortality was recorded by counting the number of unhatched eggs. The criterion for egg mortality was that the hatching line on the egg did not open up and the egg became shrivelled at the time of observation. The inhibition rate of housefly eggs was calculated using the following formula [21]; Equation (1):(1)$$Inhibition\;rate\left(\%\right)=\frac{UE}{TE}\times 100,$$
where UE is the total number of unhatched eggs, and TE is the total number of treated eggs.

The inhibition indexes (II), representing the relative efficacy of each treatment compared to α-cypermethrin was calculated using the following formula [20]; Equation (2):(2)$$II=\frac{{LT}_{50}\;or\;{LT}_{90}\;of\;\alpha –cypermethrin}{{LT}_{50}\;or\;{LT}_{90}\;of\;\;treatment},$$
where LT_50_ or LT_90_ is the 50% or 90% lethal time of α-cypermethrin or treatment against house fly eggs.

The criteria for classifying II are the following: II > 1 suggests that the treatment was more toxic to housefly eggs than α-cypermethrin; and II < 1 suggests that the treatment was less toxic to housefly eggs than α-cypermethrin; and II = 1 suggests that the treatment was equally effective to α-cypermethrin.

The increased inhibition value (IIV) was the increased efficacy in ovicidal activity of combination formulations over single-component formulations. IIV was calculated using the following formula [20], Equation (3):(3)$$IIV\%=\frac{\%\;inhibition\;rate\;of\;combination-Sum\;of\;\%\;inhibition\;rates\;of\;corresponding\;single–component\;formulations}{\%\;inhibition\;rate\;of\;combination},$$

The synergistic index (SYI) was the higher efficacy of a combination formulation over a single-component formulation of the same strength. SYI was calculated using the following formula [20,21], Equation (4):(4)$$SYI=\frac{\%\;inhibition\;rate\;of\;the\;combination}{sum\;of\;\%\;inhibition\;rates\;of\;the\;corresponding\;single–component\;formulations},$$

The criteria for classifying SYI are the following: SYI > 1 signifies that there is a synergistic effect; SYI < 1 signifies that there is an antagonistic effect, and SYI = 1 signifies that there is no synergistic nor antagonistic effect.

### 2.5. Safety Bioassay of Non-Target Aquatic Species

This study evaluated the toxicity of the tested formulations using the methods of Mounthipmalai et al. [20] and Soonwera et al. [35,36] against two non-target aquatic predators: guppy (*P*. *reticulata*) and molly (*P*. *latipinna*). Both fish predators were purchased from an organic farm in Nakhonpatom Province, Thailand (13°49′14″ N/100°03′45″ E; 57 km west of Bangkok). One hundred fish of each species were kept in a plastic container (40 × 60 × 30 cm) containing 50.0 L of clean water under the conditions of 34.0 ± 2.5 °C, 62.0 ± 2.5% RH, and 12.5 h light and 11.5 h dark periods. The starting concentration of each treatment was 10,000 ppm, following the protocol of [20,35]. Ten adult guppy or molly males were put in a plastic container (35 cm in diameter and 18 cm in height) containing 5.0 L of clean water. Each treatment was tested five times simultaneously with α-cypermethrin. Mortality rates were recorded at 1, 5, and 10 days after exposure. The LT_50_ of the treatments was compared with the LT_50_ of α-cypermethrin. The mortality rate of non-target aquatic predators was determined by the following formula [35]; Equation (5):(5)$$Mortality\;rate\left(\%\right)=\frac{D}{T}\times 100,$$
where D is the total number of dead predators, and T is the total number of treated predators.

The safety index (SI) was determined using the formula [35,36]; Equation (6):(6)$$SI=\frac{{LT}_{50}\;of\;non–target\;aquatic\;predators}{{LT}_{50}\;of\;target\;species}.$$

The criteria for classifying SI were the following: SI > 1 means that the treatment was benign to the non-target aquatic species, and SI < 1 means that the treatment was toxic to the non-target aquatic species.

The safety ratio (SR) was determined using the formula [35]; Equation (7):(7)$$SR=\frac{mortality\;rate\;against\;non–target\;aquatic\;predators\;of\;1\%\;\alpha –cypermethrin}{mortality\;rate\;against\;non–target\;aquatic\;predators\;of\;treatment}.$$

The criteria for classifying SR were the following: SR > 1 means that the treatment was safer for non-target aquatic species than 1% α-cypermethrin; SR < 1 means that the treatment was more toxic to the non-target aquatic species than 1% α-cypermethrin; and SR = 1 means that the treatment was neither safer nor more toxic than 1% α-cypermethrin.

### 2.6. Morphological Changes in Housefly Eggs after Treatment

After 48 h of an ovicidal bioassay treatment, morphological alterations, the external and internal changes in housefly eggs that underwent a treatment, were observed under a stereomicroscope (Nikon^®^ Model C-PS, Hollywood International Company Limited, Ratchathewi, Bangkok, Thailand) and photographed with a digital camera (Nikon^®^ DS-Fi2, Hollywood International Company Limited, Ratchathewi, Bangkok, Thailand) at the Research Centre, KMITL [35,38]. Scanning electron microscopy (SEM) was performed at the Scientific and Technological Research Equipment, Chulalongkorn University, Pathumwan, Bangkok, Thailand [20,36].

Regarding the SEM preparation, the egg samples were postfixed in 95% ethyl alcohol for 90 min. After that, they were dehydrated with 100% ethyl alcohol three times at 1.5 h each time. In the next step, the egg samples were dried with a CO_2_ critical point drier. The dehydrated egg samples were mounted on aluminium stubs with double-sided adhesive tape and coated with gold. All egg samples were examined with a JSM-6610LV SEM (JEOL Company Limited, Akishima, Tokyo, Japan). Photographs of egg surface morphology, an-terior and posterior poles, hatching line, plastron, islands, anastomosis, aeropyles, and micropyle were taken.

### 2.7. Ethics and Guidelines for Bioassays

The KMITL Ethics Committee approved every experiment included in this paper (under registration number KREF046703). The experimental procedures followed moral precepts and regulations regarding the usage of animals [39,40].

### 2.8. Data Analysis

All experiments were completely randomized design (CRD). The data obtained from the ovicidal and non-target bioassays were statistically analysed with IBM’s SPSS Statistical Software Package version 28 (Armonk, NY, USA). The results were reported as the inhibition rate mean ± SE (standard error) of the ovicidal bioassay and mortality mean ± SE of the non-target bioassay. One-way analysis of variance (ANOVA) was carried out by using Tukey’s test (*p* < 0.05) to investigate the differences across multiple treatment groups [41]. The time that a substance took to 50% mortality (LT_50_) and 90% mortality (LT_90_) against housefly eggs with 95% confidence limits was determined by probit analysis of mortality (number of housefly eggs that had died at 24 and 48 h after exposure). Additionally, LT_50_ with 95% confidence limits was determined by probit analysis on the mortality of two non-target species (the number of non-target aquatic predators that had died at 1, 5, and 10 days).

## 3. Results

### 3.1. Ovicidal Activity

Figure 2 shows that every single-component and combination formulation provided a high ovicidal activity with a high inhibition rate at 24 and 48 h after exposure, and the inhibition rate increased with time after exposure. After 24 h of exposure, all combination formulations were more effective (with an inhibition rate from 78.5 to 80.3%) than all single-component formulations (with an inhibition rate of 0.4 to 38.6%). Continuing along the same lines, after 48 h of exposure, all combination formulations were more effective (with an inhibition rate of 94.5 to 96.3%) than all single-component formulations (with an inhibition rate from 0.5 to 58.6%). The strongest ovicidal activity, with an inhibition rate of 94.5 to 96.2%, was achieved by two combination formulations: 1.0% star anise EO + 1.0% geranial and 1.0% lemongrass EO + 1.0% *trans*-anethole. The highest ovicidal activity among the four single-component formulations with an inhibition rate of 58.6% was achieved by 1.0% *trans*-anethole, while the lowest ovicidal activity was provided by 1.0% lemongrass EO and 1.0% star anise EO with an inhibition rate of 0.5 to 1.0%. On the other hand, 1.0% α-cypermethrin (positive control) provided a lower inhibition rate than the two combination formulations, 70.4 and 87.7% at 24 and 48 h of exposure, respectively. The negative control, drinking water, was not toxic to the housefly eggs at all, throughout the experimental period.

The LT_50_ and LT_90_ values of all treatments and 1.0% α-cypermethrin against housefly eggs and the II of all treatments are presented in Figure 3. All combination formulations exhibited high ovicidal activity with low LT_50_ and LT_90_ (LT_50_ values ranging from 17.7 to 18.6 h, and LT_90_ values ranging from 33.8 to 34.4 h), while all single-component formulations showed a high LT_50_ of 44.8 to 163.1 h and a high LT_90_ of 72.7 to 219.7 h. The 1.0% star anise EO + 1.0% geranial combination showed the lowest LT_50_ and LT_90_ values of 17.7, and 33.8 h, respectively. All of these results show a higher ovicidal activity than that of 1.0% α-cypermethrin, which had an LT_50_ of 19.8 h and an LT_90_ of 38.5 h. Regarding II, all single-component formulations were less effective than 1.0% α-cypermethrin with a low II in the range of 0.12 to 0.44 at 24 h and 0.18 to 0.53 at 48 h, while all combination formulations exhibited a high II at all experimental times (II = 1.06 to 1.14). All of them were at least 1.06 times more effective than 1.0% α-cypermethrin

Figure 4 shows the increasing inhibition values (IIVs) and synergistic indexes (SYIs) of all combination formulations against the housefly eggs when compared to the efficacy of the corresponding single-component formulations. All combination formulations exhibited high SYI throughout the whole experimental time (SYI = 1.6 to 2.4). They also exhibited a high IIV, ranging from 50.3 to 57.7% at 24 h and 37.5 to 43.7% at 48 h.

### 3.2. Toxicity to Two Non-Target Aquatic Species

The mortality rates (%) after exposures of 1, 5, and 10 days and the LT_50_ (h) values against guppies and mollies of all treatments and 1.0% α-cypermethrin are shown in Figure 5. All treatments were benign to the guppies at 1 and 5 days after exposure with a mortality rate of 0%. After 10 days of exposure, all treatments exhibited low toxicity. The mortality rate provided by all single-component formulations was in the range of 2.0 to 8.0%, with an LT_50_ of 520.8 to 607.2 h. The mortality rate provided by the combination formulations was in the range of 12.0 to 18.0% with an LT_50_ of 484.8 to 494.8 h. In contrast, 1.0% α-cypermethrin was highly toxic to the guppies with a high mortality rate of 100% at 1 day and a low LT_50_ of 0.96 h (Figure 5A). Continuing along the same line, Figure 5B shows that all treatments were benign to mollies at 1 and 5 days after exposure, exhibiting low toxicity at 10 days with a mortality rate of 4.0 to 14.0% and an LT_50_ of 492.0 to 590.4 h for single-component formulations and a mortality rate of 18.0 to 22.0% and an LT_50_ of 367.2 to 441.6 h for combination formulations. Unlike these combination formulations, 1.0% α-cypermethrin was highly toxic to mollies, with a high mortality rate of 100% at 1 day and a low LT_50_ of 0.72 h.

Figure 6A shows the safety index (SI) and the safety ratio (SR) of all treatments against guppies. The SI value of a treatment was defined as the LT_50_ against non-target species of the treatment divided by the LT_50_ against housefly eggs of the treatment. An SI value greater than one was considered safe, while an SI value less than one was considered unsafe. In this study, the SI values against the guppies of all single-component formulations (3.7 to 11.6) and combination formulations (26.6 to 27.4) was higher than one. They were considered safe for guppies. In contrast, 1.0% α-cypermethrin was highly toxic to guppies, with a low SI value of 0.04. The SR values follow a similar trend. All single-component formulations (an SR of 12.5 to 50.0) and combination formulations (an SR of 7.1 to 8.3) were 7.1 to 50.0 times safer to guppies than 1.0% α-cypermethrin. Figure 6B shows the SI and SR values of all treatments against mollies. All treatments were safe for mollies. The SI values of single-component formulations were 3.6 to 10.8. The values for combination formulations were 20.8 to 23.7. On the contrary, 1.0% α-cypermethrin was highly toxic to mollies, with an SI value of 0.05. The SR values of all treatments show that they were 4.5 to 25.0 times safer than 1.0% α-cypermethrin.

### 3.3. Morphological Changes after Ovicidal Bioassay

After 48 h of exposure to each treatment, morphological changes in the eggshells were observed under light micrograph and scanning electron micrograph (SEM), showing hatching line, plastron, aeropyles, and micropyle. The light micrographs showed morphological alterations: discolouration and damage to eggshells and hatching line, as well as dead embryos. The hatching line did not open, or it might open but the embryo inside the egg was dead after the egg had been treated, as can be seen in Figure 7. This effect was induced by 1% lemongrass, 1% star anise EO, 1% geranial, 1% *trans*-anethole, 1% lemongrass EO + 1% *trans*-anethole, and 1% star anise EO + 1% geranial. Along the same line, the SEM showed damaged, sunken, and twisted eggshells (Figure 8); damaged and swollen hatching lines and plastrons (Figure 9); abnormal islands with sunken anastomosis and closed aeropyles (Figure 10); and a swollen micropyles with closed orifices (Figure 11). All abnormal changes were induced by all treatments.

## 4. Discussion

Natural and effective insecticides for housefly management are a pressing need since houseflies have developed resistance to conventional insecticides [37,42]. In addition, most synthetic insecticides are also highly toxic to humans and the environment [43,44]. Therefore, the current global trend is to search for novel and effective alternative botanical insecticides that are gentle and safe for humans and the environment as well as highly effective against houseflies [45,46]. Single and combination formulations of EOs showed great potential as alternative natural agents and exhibited a set of diverse pest control measures that collectively contribute to pest management—ovicidal, oviposition deterrent, insecticidal, and repellent activities [19,46,47,48]. Effective natural ovicides from EOs and their main constituents are needed to reduce breeding housefly populations at the breeding site. Ovicide remains a primary prevention strategy to reduce the risk of housefly vector diseases [23,24,48,49]. Several combinations of EOs and their main constituents showed a highly synergistic ovicidal activity compared to the corresponding single components alone [37,50,51]. A desirable outcome of a synergistic combinations is that the dose or concentration of the EOs in the formulation can be reduced to obtain the same efficacy. Furthermore, dose reduction, in turn, reduces the risk of insect vector resistance [51,52,53]. The present study demonstrates that lemongrass and star anise EOs and their main compounds have great potential as sources of botanical insecticides and ovicides. They provide a high ovicidal activity against housefly eggs. All combinations in this study exhibited increased inhibition efficacy against housefly eggs with a high inhibition rate, a high II, a high SYI, and a high IIV, especially the combination of 1.0% star anise EO + 1.0% geranial. This combination produced the strongest synergistic ovicidal activity. The improvement in inhibition rate was 57.7%. Another outstanding combination was 1.0% lemongrass EO + 1.0% *trans*-anethole which gave a high level of synergy, improving the inhibition rate by 37.4%. Several studies reported that combinations of EOs and their main compounds were synergistically inhibitive against other pest insects [20,26,54]. Combinations of 2% *C*. *verum* EO + 1% geranial and 1.5% geranial + 1.5% *trans*-cinnamaldehyde provided a high inhibition rate against mosquito eggs of *Ae*. *aegypti* and *Ae*. *albopictus* (Diptera; Culicidae), with an inhibition rate of 100% and an effective inhibition rate index of 3.39 [20]. Binary combinations in equivalent mixing ratios of citral + limonene and citral+ geranyl acetate showed synergistic effects and cytotoxicity in cabbage looper ovarian cells (*Trichoplusia ni*; Lepidoptera: Noctuidae) [54]. Combinations of 5% *C*. *xanthorrhiza* EO + 5% eucalyptus EO, 10% *C*. *xanthorrhiza* EO + 10% eucalyptus EO, 5% *C*. *zedoria* EO + 5% eucalyptus EO, 10% *C*. *zedoria* EO + 10% eucalyptus EO, 5% *Z*. *zerumbet* EO + 5% eucalyptus EO, and 10% *Z*. *zerumbet* EO + 10% eucalyptus EO gave a synergistically strong ovicidal activity against human head louse eggs (*P. humanus capitus*; Phthiraptera: Pediculidae), with an inhibition rate of 96.0 to 100%, and an increased inhibition rate of more than 23.0% [26]. A combination of 10% *C*. *zedoaria* EO + 10% *E*. *globulus* EO exhibited a strong synergistic ovicidal effect against the eggs of two mosquito vectors (*Ae*. *albopictus* and *Anopheles minimus*), with an inhibition rate of 100% and an increased inhibition rate of 53.7 to 60.8% [55]. A combination of thymol + eucalyptus EO showed a high ovicidal effect against the eggs of the cattle tick, *Rhipicephalus annulatus* (Ixodida: Ixodidae) [56]. Furthermore, some combinations of monoterpenoids from EO compounds such as ϒ-terpinene + 1,8-cineole, p-cymene + ϒ-terpinene, p-cymene + 1,8-cineole, 1,8-cineole + citronellal, 1,8-cineole + linalool, linalool + citronella, (R)-pulegone + linalool, and (R)-pulegone + 1,8-cineole showed a synergistic insecticidal effect against housefly adults [51,57].

Regarding the low egg inhibition activity of all single EOs (inhibition rate of 0.5 to 1.0%), this fact is supported by a previous report [21]: lemongrass and star anise EOs showed a low inhibition rate against housefly eggs of 0.3 to 1.0%. On the other hand, 10% star anise EO exhibited a high egg inhibition rate of more than 97.0% [21]. The EOs of lemongrass and star anise at 10% showed a strong ovicidal activity against the eggs of *Ae*. *aegypti* mosquito, with an inhibition rate of 100% and an LC_50_ of 1.0 to 1.4% [58]. The lemongrass EO at 300 ppm gave a 100% inhibition rate against the eggs of the filarial mosquito (*Cx*. *quinquefasciatus*) [59]. Another article reported that ylang-ylang EO (*C*. *odorata*: Annonaceae) had a strong ovicidal activity against three mosquito vectors (*Ae*. *aegypti*, *An*. *dirus*, and *Cx*. *quinquefasciatus*), with an EC_50_ ranging from 0.5 to 1.9% [60]. EOs of anise (*P*. *anisum*), and cumin (*C*. *cyminum*) exhibited a strong ovicidal effect against two insects of stored product, the flour beetle (*Tribolium confusum*: Coleoptera) and the flour moth (*Ephestia kuehniella*: Lepidoptera) [61]. Two single-component formulations—1% geranial and 1% *trans*-anethole—in this study showed a moderate ovicidal activity against housefly eggs, with an inhibition rate of 53.2 to 58.6%. Similarly, citral (geranial + neral) and 1,8-cineole showed strong larvicidal and pupicidal activity against housefly larvae and pupae [62]. Furthermore, several monoterpenoids from EO compounds presented some ovicidal activity against eggs of other insect pests, such as carvacrol, p-cymene, and ϒ-terpinene. They provided strong ovicidal activity against cotton bollworm eggs (*Helicoverpa armigera*: Lepidoptera) [63], and geranial showed strong activity against the eggs of *Ae*. *aegypti* mosquitoes [64].

On morphological changes to eggs after the ovicidal assay, all treatments induced damage and abnormality—sunken and twisted eggshells, swollen hatching line, hatching line has never opened, or it has opened but the embryo inside the egg was dead, and damage of plastron with swollen anastomosis and closed aeropyles. In addition, the micropyle was closed and covered with a thick layer of substance. Similarly, rice bran acetonic extract (*Oryza sativa*) and chitin synthesis inhibitor induced ultrastructural changes in housefly eggs that caused embryonic mortality, abnormal or blocked embryonic development [65]. D-limonene, geranial, and *trans*-cinnamaldehyde induced egg morphological changes in *Ae*. *aegypti* and *Ae*. *albopictus*, causing damage to the exochorionic, abnormal shape of cell borders, papillates, and aeropyles. Furthermore, the aeropyles were covered with layers of EO, blocking the respiratory system, resulting in unhatched eggs with dead embryos inside [20]. Other studies reported that a combination of EOs, *C*. *zedoaria* and *E*. *globulus*, blocked the aeropyles of lice eggs, causing unhatched eggs and embryo mortality inside the eggs [26].

Regarding the mechanism of the ovicidal action of single-component and combination formulations, from the evidence of the morphological changes discussed above, the formulations exerted their actions through multiple mechanisms. They blocked the hatching line and plastron, causing damage to the respiratory and nervous systems. Geranial acted toxically on the neurological system by penetrating the embryonic cuticles and disturbing the embryogenesis development [64,65]. Lemongrass EO affects the biochemical activities in housefly larvae, reducing the production of proteins, lipids, and amylase. It inhibited the enzymes butyrylcholinesterase (BChE) and acetylcholinesterase (AChE) [66,67]. Star anise EO reduced AChE activity and progeny production of the corn weevil (*Sitophilus zemais*; Coleoptera: Curculionidae) [68]. Trans-anethole inhibited the AChE activity of mediterranean flour moth (*Ephestia kuehniella*; Lepidoptera: Pyralidae) [69]. Additionally, a combination of monoterpenes was a more effective AChE inhibitor of the mosquito nervous system than a single monoterpene [70].

Several EOs and their main components are largely presented as safe substances, eco-friendly, and non-hazardous to non-target species such as natural enemies, predators, parasitoids, and pollinators [35,36,71]. In this study, all treatments were safe for non-target aquatic predators (guppy and molly), with high SI and a high SR when compared to 1% α-cypermethrin. Two combinations—1% star anise EO + 1.0% geranial and 1.0% lemongrass EO + 1.0% *trans*-anethole—were outstandingly non-hazardous to both aquatic predators, with an SI of 20.4 to 20.5 (guppy) and 26.6 to 27.4 (molly). This finding agrees with a previous report that combinations of cinnamon EO + geranial (2:1 ratio), lemongrass EO + D-limonene (2:1 ratio), and citrus EO + geranial (2:1 ratio) were safe for non-target species, molly and guppy, with a high LC_50_ of 4415.4 to 5921.3 ppm and a high biosafety index (BI) of 1.03 to 2.57 [20]. Other articles report that combinations of 1.25% lemongrass EO + 1.25% star anise EO and 0.25% geranial + 0.25% *trans*-anethole were safe for molly and guppy with a low mortality rate (less than 10%) and a high safety index, more than 260 [36,37]. Some single monoterpenes such as thymol were fairly toxic to guppy with an LC_50_ value of 10.99 to 12.51 mg/L, while 1,8-cineole was less toxic to guppy with an LC_50_ value of 1701.93 to 3997.07 mg/L [72]. Conversely, 1.0% α-cypermethrin was highly toxic to both guppy and molly with a high mortality rate of 100% at 1 day and a low LT_50_ of 0.72 to 0.96 h. Cypermethrin was reported to be toxic to several non-target species, including guppy and molly, other natural enemies of insects, predators, and pollinators [73,74]. Cypermethrin is hazardous to the nervous system of humans and other mammals. It is poisonous to neuron cells and neurotransmitter receptors, causing nausea, incontinence, hyperexcitation, shortness of breath, and even death [75,76]. On the other hand, single–component and combination formulations of EOs and their main constituents were safe and benign to the non-target species. A testament to its safety is that lemongrass and star anise EOs have long been used as traditional medicine, food preservatives, and food ingredients of Asians since ancient times. They neither damaged nor altered human cells [77,78]. All combination formulations in this study exhibited a stronger inhibition efficacy against housefly eggs than single-component formulations. The combination formulations were so toxic to the housefly eggs that the eggs were destroyed so quickly that the embryo died inside the eggs and hence did not hatch and develop further.

## 5. Conclusions

Two outstanding combination formulations of 1.0% star anise EO + 1.0% *trans*-anethole and 1.0% lemongrass EO + 1.0% *trans*-anethole exhibited a highly synergistic ovicidal activity and a highly increased inhibition value against housefly eggs at low concentrations and were safe to two non-target species. They are readily available, sustainable, safer, and more effective than cypermethrin, the popular synthetic housefly insecticide. They should be further developed into a natural ovicidal agent for managing housefly populations in houses and farms. Future studies would include an investigation confirming or negating that several different modes of action are at work to produce the strong efficacy. Ultimately, bio-efficacy experiments of these two combinations need to be conducted. The combinations should be formulated in the form of a spray against housefly adults and the form of an aqueous solution against the eggs, larvae, and pupae stages of houseflies in urban areas and farms. Moreover, toxicity studies against non-target predators, parasites, pollinators, and earthworms should be conducted.

## Figures and Tables

**Figure 1 insects-15-00481-f001:**
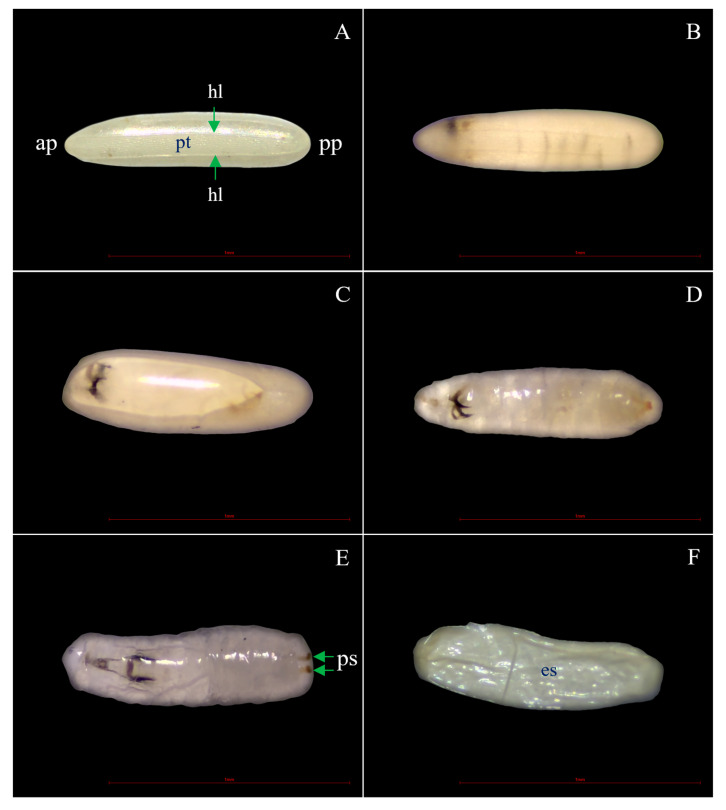
Light micrographs of housefly eggs: (**A**) dorsal view of a fertile egg aged 2–3 h showing the anterior pole (ap) on the left, the posterior pole (pp) on the right, the plastron (pt) on the median and the hatching line (hl) (green arrow); (**B**) a fertile egg aged 6 h; (**C**) a fertile egg aged 12 h; (**D**,**E**) the first instar larvae with posterior spiracles (ps) (green arrow); and (**F**) eggshell (es).

**Figure 2 insects-15-00481-f002:**
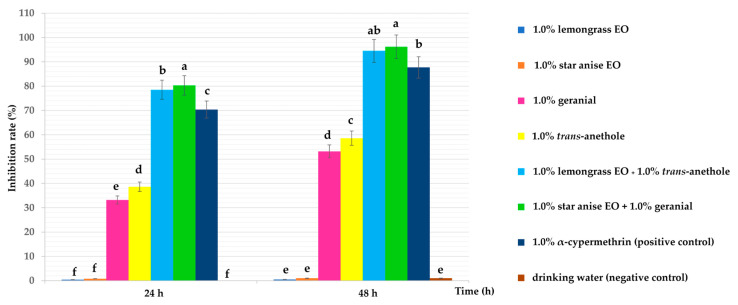
Inhibition rate (%) ± SE versus exposure time of treatments, 1% α-cypermethrin, and drinking water against housefly eggs. Note: The mean inhibition rates within a column followed by a different letter differ significantly according to the Tukey’s test at *p* < 0.05.

**Figure 3 insects-15-00481-f003:**
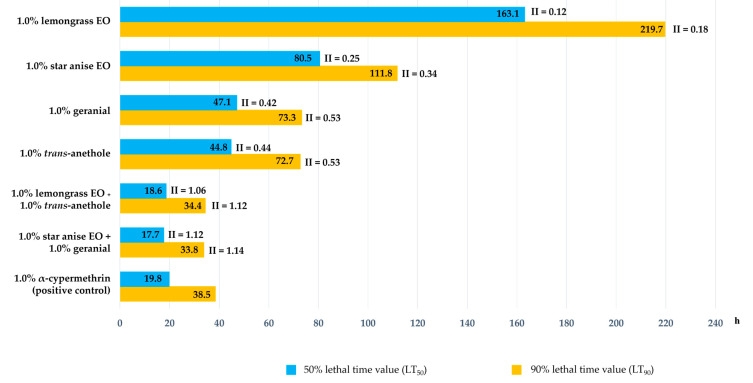
The 50% lethal time (LT_50_) and 90% lethal time (LT_90_) of treatments and 1% α-cypermethrin against housefly eggs, and inhibition index (II) comparing the strength of treatments versus 1% α-cypermethrin. Note: II > 1 suggests that the treatment was more toxic to housefly eggs than α-cypermethrin; II < 1 suggests that the treatment was less toxic to housefly eggs than α-cypermethrin; and II = 1 suggests that the treatment was equally effective to α-cypermethrin.

**Figure 4 insects-15-00481-f004:**
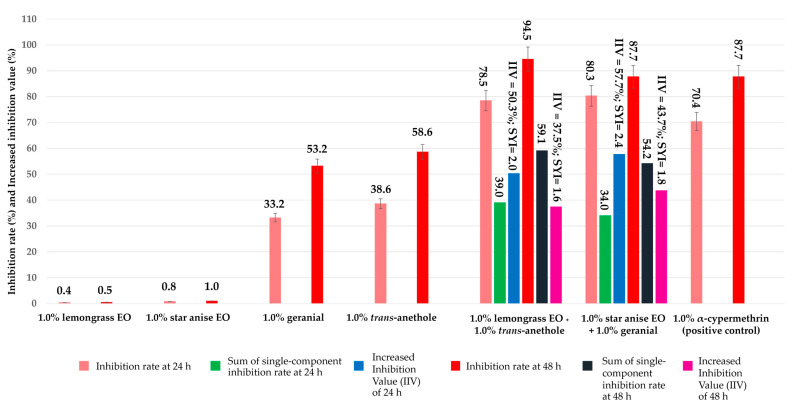
Increased inhibition value (IIV) and synergistic index (SYI) of two combinations compared to the corresponding single-component formulations. Note: SYI > 1 signifies that there is a synergistic effect; SYI < 1 signifies that there is an antagonistic effect; and SYI = 1 signifies that there is no synergistic nor antagonistic effect.

**Figure 5 insects-15-00481-f005:**
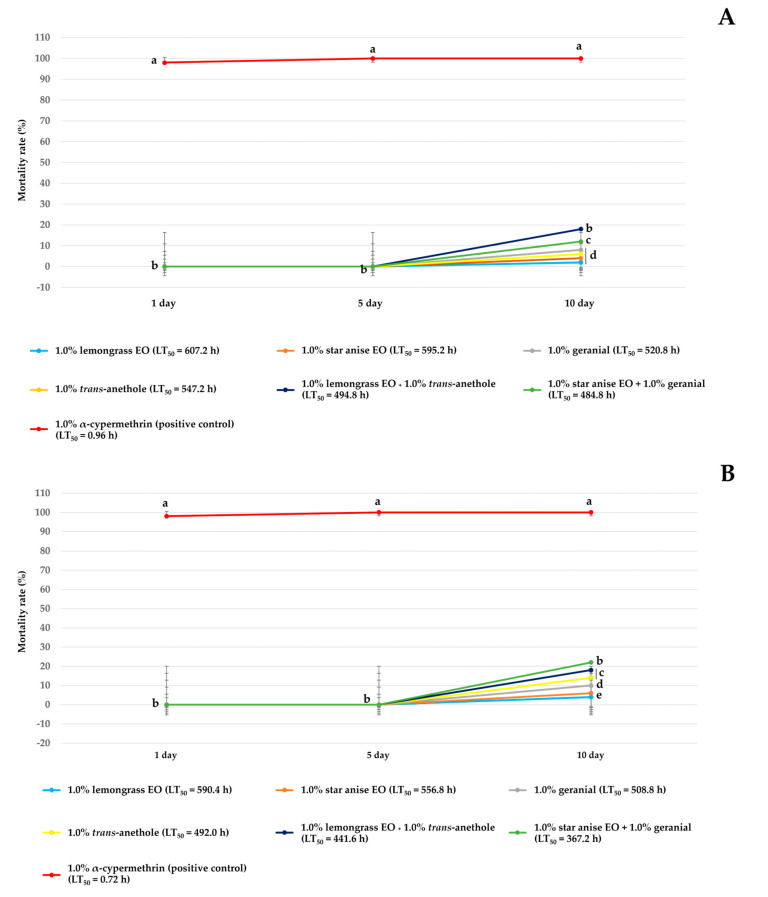
Mortality rate (%) ± SE and LT_50_ of single-component and combination formulations and 1% α-cypermethrin against non-target species: guppy (**A**) and molly (**B**). Note: Mean mortality rates within a column (1, 5, and 10 days) followed by a different letter differ significantly by Tukey’s test at *p* < 0.05.

**Figure 6 insects-15-00481-f006:**
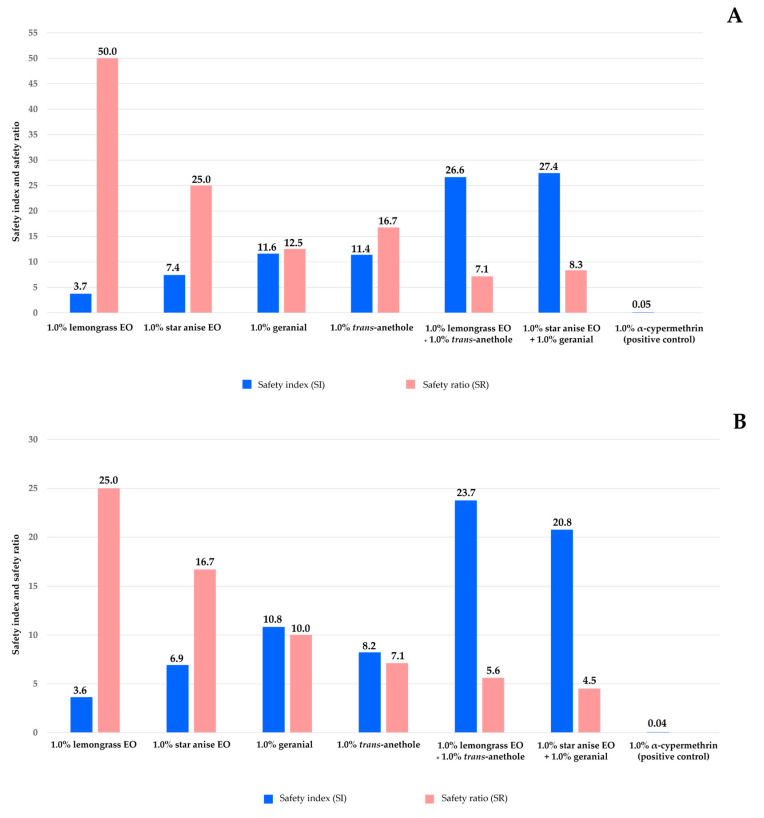
Safety index (SI) and safety ratio (SR) values of all treatments and 1% α-cypermethrin against non-target species: guppy (**A**) and molly (**B**). Note: An SI value greater than one was considered safe, while an SI value less than one was considered unsafe. SR > 1 means that the treatment was safer for non-target aquatic species than 1% α-cypermethrin; SR < 1 means that the treatment was more toxic to the non-target aquatic species than 1% α-cypermethrin; and SR = 1 means that the treatment was neither safer nor more toxic than 1% α-cypermethrin.

**Figure 7 insects-15-00481-f007:**
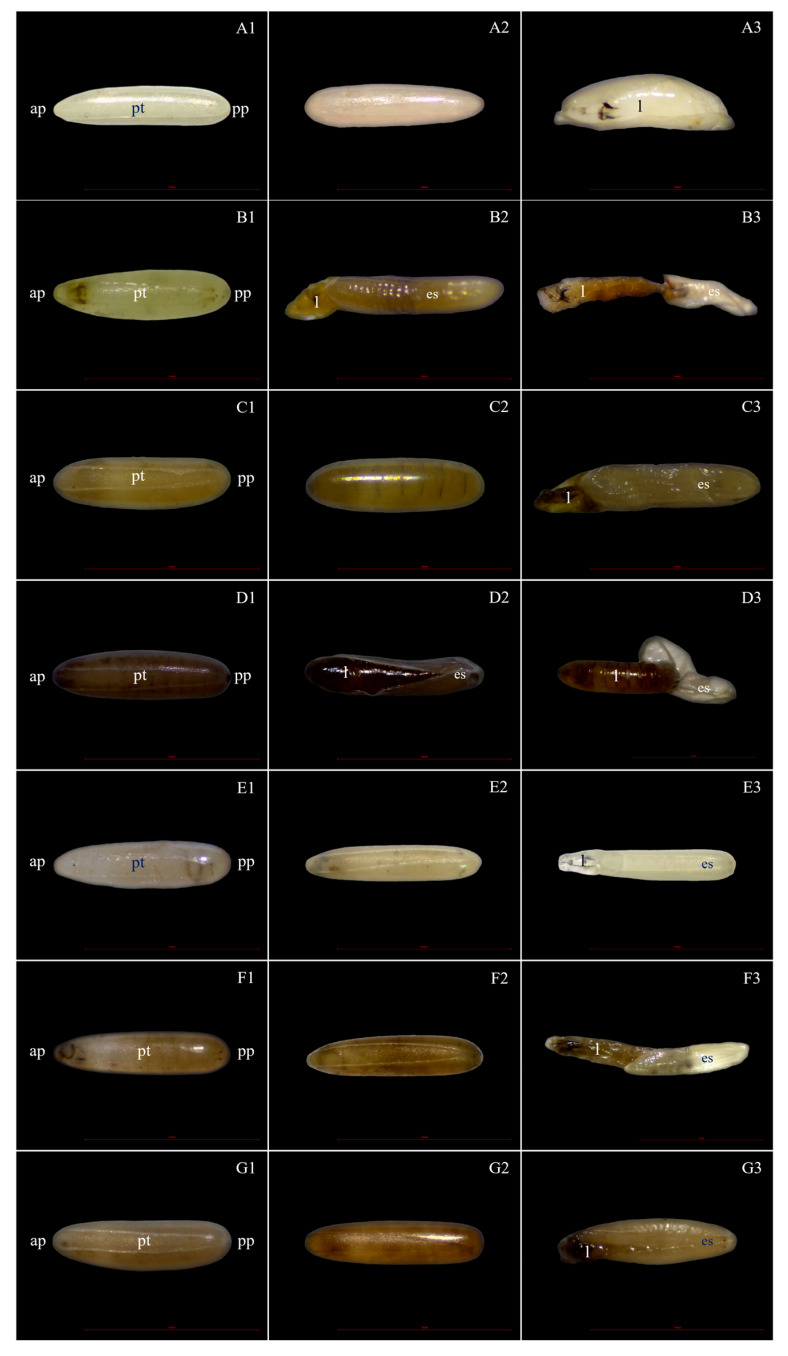
(**A1**,**A2**) Light micrographs of housefly eggs showing a complete fertile egg; (**A3**) the first instar larva (l) and morphological changes induced by treatment, showing a damaged embryo and dead embryo inside the egg after being treated with 1% lemongrass EO (**B1**–**B3**), 1% star anise EO (**C1**–**C3**), 1% geranial (**D1**–**D3**), 1% *trans*-anethole (**E1**–**E3**), 1% lemongrass EO + 1% *trans*-anethole (**F1**–**F3**), and 1% star anise EO + 1% geranial (**G1**–**G3**). Note: Anterior pole (ap), posterior pole (pp), plastrons (pt), and eggshell (es).

**Figure 8 insects-15-00481-f008:**
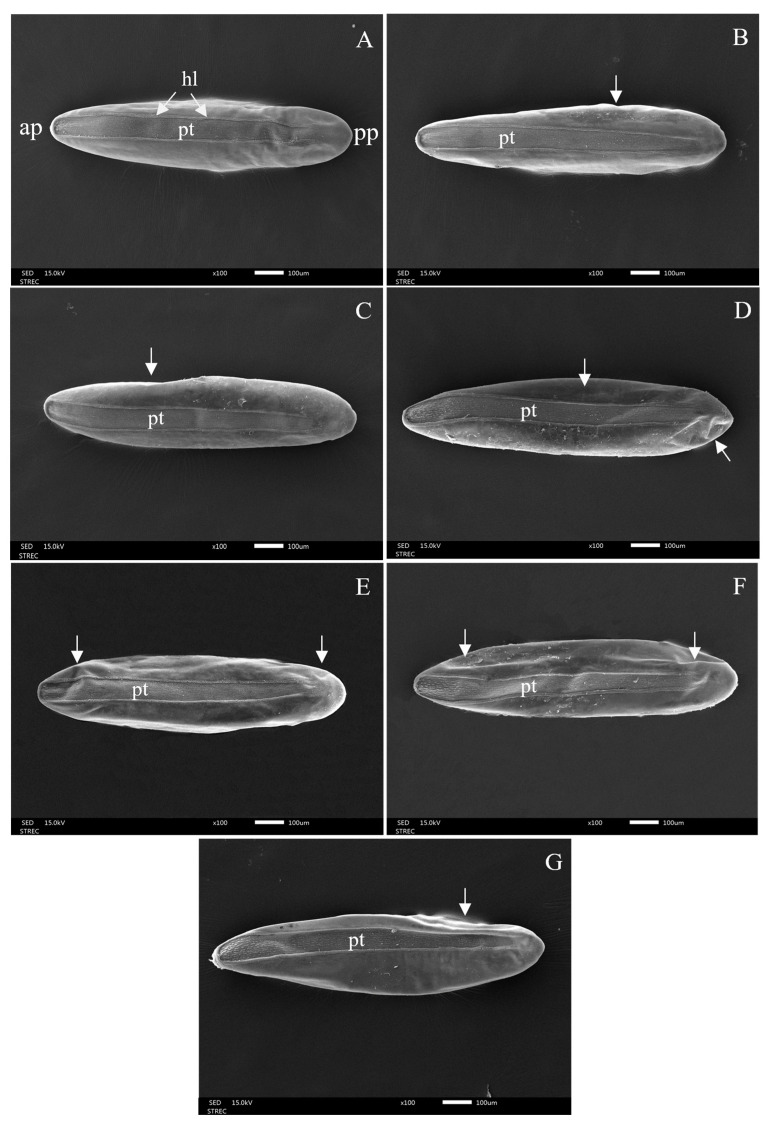
Scanning electron micrographs of the dorsal view of a housefly egg showing the normal egg surface with anterior pole (ap) on the left, posterior pole (pp) on the right, plastrons (pt), hatching line (hl) (white arrow) on the median of the untreated egg (**A**), and morphological changes induced by treatment, showing damages and abnormality of egg surfaces after the egg was treated with 1% lemongrass EO (**B**), 1% geranial (**C**), 1% star anise EO (**D**), 1% *trans*-anethole (**E**), 1% lemongrass EO + 1% *trans*-anethole (**F**), and 1% star anise EO + 1% geranial (**G**).

**Figure 9 insects-15-00481-f009:**
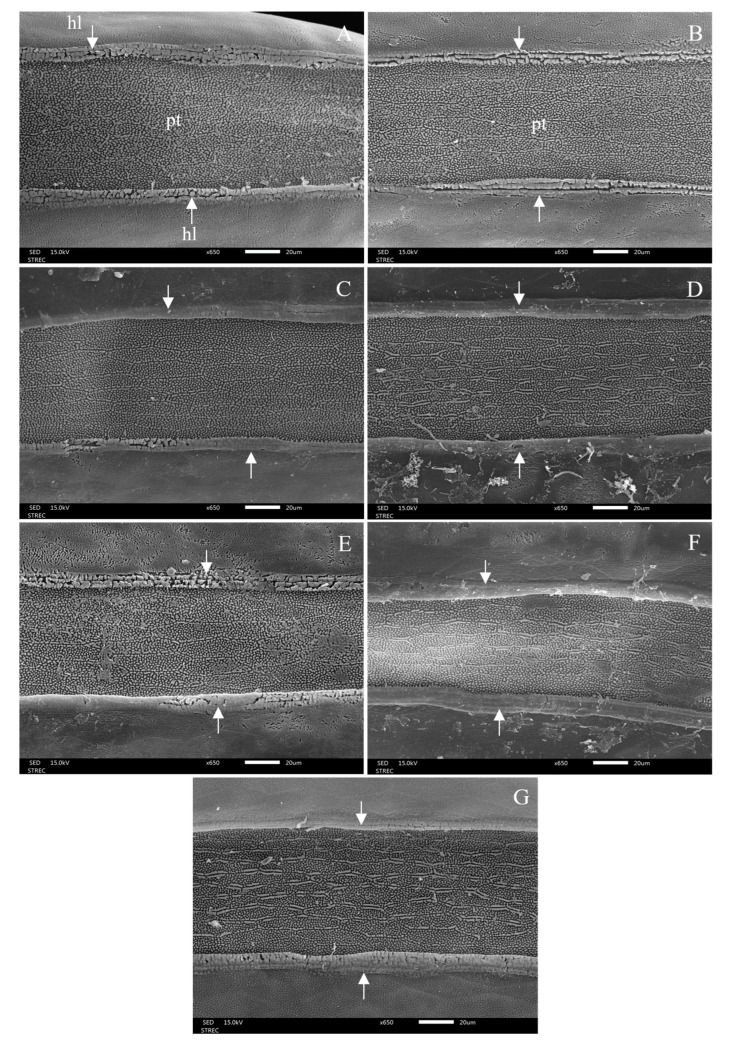
Scanning electron micrographs of the median area of a housefly egg showing the normal hatching line (hl) (white arrow) and plastron (pt) of an untreated egg (**A**). They also show an abnormal and swollen hatching line and a damaged plastron after the egg was treated with 1% lemongrass EO (**B**), 1% geranial (**C**), 1% star anise EO (**D**), 1% *trans*-anethole (**E**), 1% lemongrass EO + 1% *trans*-anethole (**F**), and 1% star anise EO + 1% geranial (**G**).

**Figure 10 insects-15-00481-f010:**
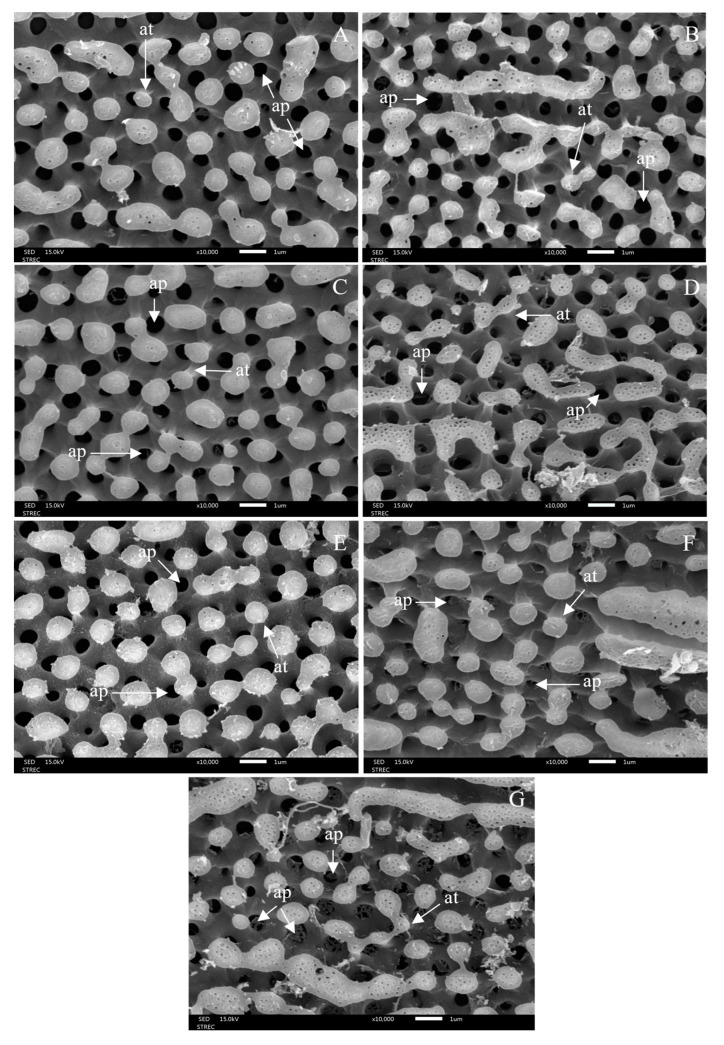
Scanning electron micrographs of the plastron of an untreated egg in closer view, showing normal islands with normal anastomosis (at) and aeropyles (ap) (white arrow) (**A**); damaged and swollen anastomosis and closed aeropyles after the egg was treated with 1% lemongrass EO (**B**), 1% geranial (**C**), 1% star anise EO (**D**), 1% *trans*-anethole (**E**), 1% lemongrass EO + 1% *trans*-anethole (**F**), and 1% star anise EO + 1% geranial (**G**).

**Figure 11 insects-15-00481-f011:**
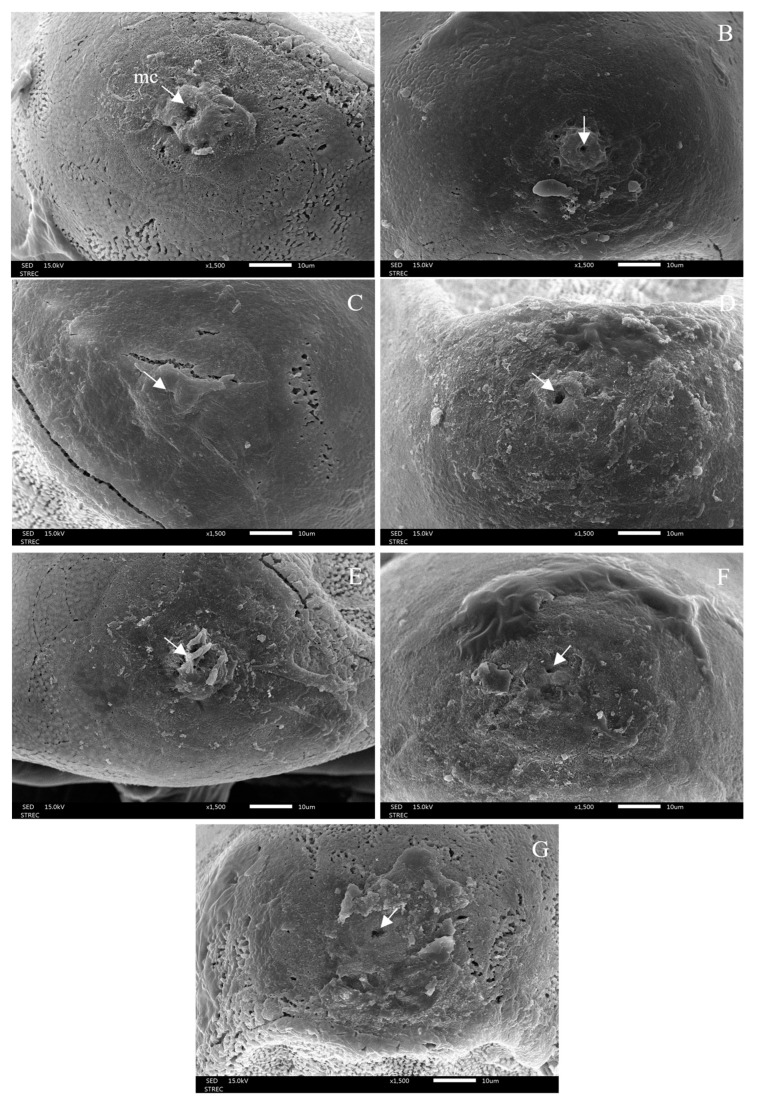
Scanning electron micrographs of a micropyle (mc) (white arrow) in the anterior region with smooth ornamentation and a single orifice in the middle of an untreated egg (**A**); an abnormal micropyle with rough ornamentation with swollen and closed orifice after the egg was treated with 1% lemongrass EO (**B**); 1% geranial (**C**); 1% star anise EO (**D**); 1% *trans*-anethole (**E**); 1% lemongrass EO + 1% *trans*-anethole (**F**); and 1% star anise EO + 1% geranial (**G**).

## Data Availability

All relevant data are included in the article.

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
