# Peer review of "Ovicidal Toxicity and Morphological Changes in Housefly Eggs Induced by the Essential Oils of Star Anise and Lemongrass and Their Main Constituents"

_insects, 2024, doi:10.3390/insects15070481_

Round 1

Reviewer 1 Report

Comments and Suggestions for Authors

As I stasted in my points on the work done in the manuscript, the authors made an excellent evaluation design in the laboratory, but using different materials as f fly eggs and mosquitoes egg, given that Aedes aegypti is a is the most important vector of many diseases and now it is almost fashionable to talk about Aedes aegypti in research focused on solutions to control , there is mention of ovicidal also using the Aedes's eggs.

and consequently present it as a product to contain and control the vector of diseades, even the authors did not say it. Ovicidal product against misquotes is not appropriate control measure and who work on it is a waist of time ok for fly control but not for mosquito', s egg. It cannot be affirmed that the eggs used for lab evaluation and the result is good so it works for control A.aegypti too. The aurhors should mention clearly that the A. Aegypti's eggs were just live material on which were carried out the evaluation.

To be a valid product they have to evaluate in fileld too.
Therefore, like many products, essential oil products show good performaces in lab but not in the field. We see that scientists start working and finding solutions in the laboratory which in the end do not become a useful product to contain the problem.

When you look at the result, everything goes well." Two observations:

1.  this work is performed in an excellent evaluation design in the laboratory and deserves approval for how it was conducted such as intent, materials and methods and effectiveness result in the laboratory. There are no elements such as an in-field solution or a product to be used as an alternative to another product.

Mention of Aedes agypti must not be alluded to as a product to contain the abundance of mosquitoes. Far from being mentioned for the approach to vector control.

2.  like all natural products it is necessary to say which standard for any large-scale product, and possibly to write on a label as a product and to be used as a finished product

Behind an excellent work it must lead to use in the field and its usefulness in the field and not only in the laboratory

Explain that use  eggs of Aedes aegypti is just a material and clarify that the use of formulation is not mentioning at all to vector control of mosquito

Reviewer 2 Report

Comments and Suggestions for Authors

This research demonstrated the efficacy of combined treatment with essential oil on houseflies, providing a safer and more effective method compared to α-cypermethrin, a conventional pesiticide. Specifically, it showed the ovicidal effect on houseflies with high-quality images of eggs.

Please see the attached file for my comments.

Reviewer 3 Report

Comments and Suggestions for Authors

The manuscript investigates the ovicidal effect of lemongrass (Cymbopogon citratus) EO + trans-anethole and star anise (Illicium verum) EO + geranial against house fly eggs. The authors found these compounds were effective against eggs and caused embryonic damage and mortality of the treated eggs. The authors studied these effects on the morphological changes in the treated eggs by stereo microscope and scanning microscopy. Moreover, the authors studied the safety of these compounds on two free-living organisms and they found that these natural products are safe in comparison with chemical insecticide.

In my view, even though this study is robust and properly conducted, but its significant application in the field is nill. In the control strategy, we must use a dose killer for all stages, but to study the effectiveness of these compounds against eggs only is non-practical work.   

Round 2

Reviewer 2 Report

Comments and Suggestions for Authors

No more comments.

The mistakes have been corrected.